# The Role of Cryoprotective Agents in Liposome Stabilization and Preservation

**DOI:** 10.3390/ijms232012487

**Published:** 2022-10-18

**Authors:** George Frimpong Boafo, Kosheli Thapa Magar, Marlene Davis Ekpo, Wang Qian, Songwen Tan, Chuanpin Chen

**Affiliations:** 1Xiangya School of Pharmaceutical Sciences, Central South University, Changsha 410013, China; 2Department of Pharmaceutics, School of Pharmacy, China Pharmaceutical University, Nanjing 211198, China

**Keywords:** freeze drying, freeze-thaw, liposome(s), cryoprotectants, drug delivery, phospholipid–CPA interactions

## Abstract

To improve liposomes’ usage as drug delivery vehicles, cryoprotectants can be utilized to prevent constituent leakage and liposome instability. Cryoprotective agents (CPAs) or cryoprotectants can protect liposomes from the mechanical stress of ice by vitrifying at a specific temperature, which forms a glassy matrix. The majority of studies on cryoprotectants demonstrate that as the concentration of the cryoprotectant is increased, the liposomal stability improves, resulting in decreased aggregation. The effectiveness of CPAs in maintaining liposome stability in the aqueous state essentially depends on a complex interaction between protectants and bilayer composition. Furthermore, different types of CPAs have distinct effective mechanisms of action; therefore, the combination of several cryoprotectants may be beneficial and novel attributed to the synergistic actions of the CPAs. In this review, we discuss the use of liposomes as drug delivery vehicles, phospholipid–CPA interactions, their thermotropic behavior during freezing, types of CPA and their mechanism for preventing leakage of drugs from liposomes.

## 1. Introduction

Liposomes undergo freezing during formulation and post-formulation in the form of freeze-drying and freeze-thawing respectively. These techniques commonly employ the use of cryoprotective agents (CPAs) or cryoprotectants to warrant the stability of liposomes during the freezing process [1]. The major obstacles when it comes to freezing lipid membranes are intracellular formation of ice, cryoinjuries and osmotic injuries during the freeze-thaw cycle [2,3]. CPAs prevent damage by regulating the rates of water transport, nucleation and ice formation [2,4]. Examples of CPAs frequently used when freezing liposomes are dimethyl sulfoxide (DMSO), glycerol, polymers (polyampholytes), ethylene glycol, propylene glycol, sugars such as trehalose and sucrose, amongst others.

The inception of liposomes as drug carriers have been a big game changer in the pharmaceutical space due to their numerous drug delivery abilities. Due to their structural resemblance to biologic membranes and their utility as drug delivery systems, liposomes have been the subject of research since the 1960s [5]. Unlike some drug carriers, liposomal drug delivery systems (DDS) are non-cytotoxic with enhanced stability and high encapsulation efficiency [6,7]. Liposomes also increase the therapeutic index of drugs by increasing the half-life of drugs, enabling active targeting via site specific ligands and improving the transport of drugs across membranes [8,9]. Furthermore, the great biocompatibility [10] and the potential of liposomes to entrap hydrophobic drugs in their bilayer membrane and hydrophilic drugs in their aqueous core makes them more desirable in the pharmaceutical industry [11].

In spite of all the tremendous characteristics of liposomes, there are still some concerns about their prolonged storage stability and preservation. Additionally, freezing of liposomes for lengthy liposomal clinical trials destabilizes the liposomes without the use of cryoprotectants [12] or the use of ineffective CPAs (Figure 1). The majority of studies conducted on the preservation of liposomes is mainly focused on lyophilization (freeze drying) [13,14,15]. Lyophilization is typically used as an essential method to increase the stability of liposomal drugs, make storage, transportation, and product shelf-life easier, and all of these goals. Moreso, even though the use of non-permeating CPAs (carbohydrates and sugars) have been largely employed in the lyophilization of liposomes [16,17,18], little information is available on the use of other permeating CPAs besides DMSO in the lyophilization of liposomes [19,20]. Therefore, in this paper, we seek to review the application of liposomes as drug delivery systems and lipid–CPA interactions of phospholipids as well as their thermotropic behavior. Then, we further discuss the various types and examples of CPAs that can be used during liposome freezing and how lipid interactions with CPAs prevent leakage of drugs from liposomes.

## 2. Application of Liposomes in Drug Delivery

Among other drug carriers, liposomes exhibit limitless capacities for effective delivery of drugs to the targeted area [6]. In addition to boosting stability by encapsulation, liposomes have also been seen to increase pharmacological efficacy and therapeutic index, improve pharmacokinetic effects (i.e., decrease clearance and lengthen circulation lifetime) and reduce the toxicity of encapsulated substances [6,8,21]. Liposomes were first described as swollen phospholipid bodies with absolute size and shape variability that completely encloses an aqueous mass inside of it to form little sphere-shaped vesicles (Figure 2) [22]. The key components used in the preparation of liposomes are cholesterol, glycolipids, sphingolipids, non-toxic surfactants, long-chain fatty acids and membrane-bound proteins [23]. Liposomes can be synthesized in a variety of ways for drug delivery. Nevertheless, there are four fundamental steps in each of the procedures used to fabricate liposomes. These include drying lipids from organic solvents, dispersing them in an aqueous media, purifying the resulting liposomes and evaluating the finished product. Both passive and active loading methods are applied when loading a drug into a liposome. The passive loading methodology is known to include three different methods: solvent dispersion method, mechanical dispersion method, and detergent removal method (i.e., to remove free drug) [24,25].

Liposomes can differ in their design based on their preparation method (extrusion techniques, reverse phase evaporation method, sonication, dehydration method, and more recently, microfluidics) [26,27] and their structural parameters in terms of size, charge and lamellarity (unilamellar, multilamellar and oligolamellar vesicles) [6]. They can also differ based on their function, application and composition. Some examples and types of liposomes based on their composition are conventional liposomes, long-circulating liposomes, temperature-sensitive liposomes, pH-sensitive liposomes, light-sensitive liposomes, immunoliposomes (ILs), enzyme-sensitive liposomes and magnetic-response liposomes (Figure 2) [6,28]. A liposome can range from nano to micro-sized vesicles having one or two membrane layers.

A key factor influencing the circulatory half-life of a liposome is the size of its vesicle. The ability of a liposome to encapsulate active agents is influenced by both the number and size of bilayers. Liposomes are divided into three types based on the quantity and size of their bilayers: unilamellar vesicles (ULV), multilamellar vesicles (MLV) and multivesicular vesicles (MVV) (Figure 3) [29,30]. Unilamellar vesicles can also be divided into giant unilamellar vesicles (GULV), large unilamellar vesicles (LUV) and small unilamellar vesicles (SUV), all of which have a size range between 100 and 1000 nm. While the vesicles in multilamellar liposomes have an onion structure, the aqueous solution is entrapped by a single spherical phospholipid bilayer in a unilamellar liposome.

### 2.1. Liposome-Cell Interaction

Liposomes interact with cells in different ways to exert their effects inside the human body. There are two basic forms of liposome interactions with cells: pH-sensitive and cationic interactions [31]. Negatively charged liposomes entrap DNA within their aqueous compartment in a pH-sensitive interaction, as opposed to forming stable complexes (lipoplexes). Whereas, in a cationic reaction, liposomes made up of positively charged lipids (lipofectin) and co-lipids, interact with negatively charged DNA molecules to produce a stable complex [32]. pH-sensitive interaction with cells is usually applied during DNA delivery in vivo, while gene therapy makes use of cationic interactions. Moreover, when cells and liposomes interact, the following occur, as exhibited in Figure 4 [33]: (1) Liposomes adhere to cellular membranes and appear to fuse with them, releasing their contents into the cell; (2) Liposomes are engulfed by cells, incorporating their phospholipids into the cell membrane and releasing the constituents; (3) Liposomes are taken up by phagocytic cells, where the lysosomes break down their phospholipid walls and release their constituents.

Another way liposomes are internalized by cells is through the endocytic pathway. For entrapped molecules to be delivered intracellularly, liposomes interact with endosomes and lysosomes and require membrane fusion or lipid mixing with these membranes. The most common endocytotic mechanisms for the intracellular delivery of liposomes in normal cells are caveolae-mediated endocytosis (CavME), clathrin-mediated endocytosis (CME) and micropinocytosis [34,35]. CavME tends to internalize relatively small lipoplexes, while CME and macropinocytosis prefer to take up the larger ones by dendritic cells (DCs) originating from bone marrow [36]. However, Bae et al. discovered that in COS-7 cells, 1,2-dioleoyl-3-trimethylammonium propane (DOTAP) liposomes of various diameters and cholesterol ester liposomes were internalized mostly by the CME pathway [37].

Furthermore, to achieve the desired efficacy of nucleic acid-loaded liposomal therapy, endosomal escape, which is dependent on membrane fusion, is an essential step for lipoplexes to be trapped in endosomes. According to numerous studies, liposomes can transport drugs that are entrapped into the cytosol directly by membrane fusion as opposed to through the endocytic pathway [38,39,40,41,42]. The intracellular pharmacokinetics of liposomal compositions and their absorption processes have been shown to be correlated. There are special combinations of liposomes that have various absorption mechanisms but the same lipid makeup. For instance, cationic liposomes (K3C14) with lysine heads and ditetradecyl tail chains demonstrated notable cellular uptake and lysosome disruption while maintaining the activity of the compounds they contained [43,44]. Cationic liposomes constructed of K3C16, however, demonstrated cellular internalization through a membrane fusion mechanism [45].

### 2.2. Applications of Liposome

Most drugs, irrespective of their solubility, can be encapsulated in a liposome, and such drugs are protected from the action of external media, particularly enzymes [46,47] and inhibitors. However, for a liposome to be considered an effective drug delivery system, understanding the lipid–drug interaction and the liposome disposition mechanism is very necessary. Usually, drugs with widely ranging lipophilicities are encapsulated in a liposome either in the aqueous core or at the bilayer interface [46]. Drugs encapsulated in liposomes are expected to be transported without rapid degradation and minimum side effects to the recipients, as liposomes possess properties of weak immunogenicity, limited intrinsic toxicity, produce no antigenic or pyrogenic reactions and are biologically inert [46].

The success of liposomes as drug carriers has been reflected in a number of liposome-mediated formulations that are currently available commercially and approved for use in clinical studies [48]. A sizable number of other anti-cancer medicines, including DaunoXome^®^, Depocyt^®^, Myocet and OnivydeTM [48,49], have been successfully produced since the first liposome formulation (Doxil^®^) was developed [50]. Furthermore, the use of liposomes is not only limited to the administration of anti-cancer therapies but also anti-bacterial, nucleic acids, anti-viral (Epaxal^®^, Inflexal^®^), pain relief agents (DepoDur^TM^, Exparel^®^) and anti-fungal (Abelcet^®^, Ambisome^®^, Amphotec^®^) [40,41,42,51,52,53], as exhibited in Table 1.

Liposomes have been applied in a number of purposes due to the many advantages they possess. As per their applications, liposomes have been useful both generally and clinically [42]. A general application of a liposome can be ascribed to its use in brain targeting. This is evidenced by the fact that, due to the biodegradable and biocompatible nature of liposomes, liposomal amitriptyline when administered was found to be able to cross the blood brain barrier rather than when administered systemically, proving its application in brain targeting [44]. As seen in Table 1, liposomes have generally been applied also in respiratory disorders, vaccine adjuvants and anti-infective agents, accounting for their pleiotropic potential [54] and eye disorders, i.e., recently approved liposomal verteporfin was found to be effective against eye disorders [55].

The application of liposomes in the delivery of NA therapeutics has helped overcome the challenge of clearance by enzymatic degradation nucleuses and the reticuloendothelial system (RES). As such, liposomes are now employed as carriers for the delivery of either siRNAs, antisense ODNs/ON, ribozymes, or used as plasmid vectors for gene therapy with the goal of downregulating certain genes [56,57]. Genetic modification and low molecular therapy can be combined in order to boost the efficacy of gene therapy. To increase the anti-cancer efficacy of doxorubicin (DXR) in lung cancer cells, Saad et al. [58] developed cationic liposomes for the simultaneous delivery of DXR and siRNA, targeting the multi-drug resistance (MDR) protein. Later, Peng et al. [59] evaluated the efficiency of a novel thermosensitive magnetic liposome for the simultaneous delivery of SATB1 and DXR short hairpin RNA (shRNA) to gastric cancer cells. In comparison to individual delivery, it was demonstrated that DXR and SATB1 shRNA were delivered into MKN-28 cells, a human gastric adenocarcinoma, with increased drug delivery efficacy and high gene transfections. This resulted in growth inhibition in gastric cancer cells in both cell and animal models.

Due to the rapid development of nanomedicine, liposome-mediated protein delivery nano-formulations have recently been developed for clinical usage. Protein-repellent polymers, such as polyethylene glycol (PEG), can be used to modify liposome surfaces in order to get around the problems with protein transport that have been addressed [60,61]. Entrapping proteins into liposomes can increase the durability of protein therapies because the lipid bilayer shields them from deterioration. Additionally, active ligands can alter PEGylated liposomes to increase active targeting and to prolong in vivo circulation [62]. For example, for effective insulin delivery, protein corona liposomes (PcCLs) were designed in which insulin was encapsulated within the CLs. According to this study, PcCLs’ hydrophilic and neutral-charge protein corona can effectively penetrate mucus and progressively dissolve through the action of enzymes. To enhance insulin delivery over the transepithelial membrane, these PcCLs may break down. The evidence was convincing about the enhanced transepithelial permeability and cellular absorption of PcCLs. In type I diabetic rats treated with PcCLs, lower blood glucose levels and greater oral bioavailability of up to 11.9% were seen [63]. Therefore, PcCLs are a newly developed technique for administering peptide/protein medications to lessen GI-related problems.

Furthermore, liposomes have become an appealing delivery method for antibodies and enzymes due to their biocompatibility, biodegradability and controlled release qualities. Considerable work has gone into creating liposomes that can be specifically targeted by different ligands and antibodies [64]. The idea of ILs was created to combine the effects of antibodies with liposomes. Gregoriadis et al. [65] looked at the use of IgGs made against various cells and discovered that they could specifically swallow liposomes. According to Leserman et al. [66], liposomes with antibodies attached to their surfaces made specific contact with target cells.

In addition, the use of liposome as a vaccine carrier has lately gained attention after years of use as a medication carrier. Due to liposomes’ physicochemical characteristics, tolerance by the human body, low cytotoxicity and chemical and structural flexibility, they have been chosen as vaccine delivery vehicles over other types. A liposome’s ability to contain a hydrophilic antigen, adjuvant or lipophilic component that can move between bilayer lipids is referred to as chemical flexibility. It is possible to conjugate hydrophilic antigens on their surface, which increases antigen accessibility and encourages phagocytic absorption. The ability to alter liposomal characteristics by varying lipid concentration is associated to structural flexibility. For instance, various cationic lipid compositions have frequently been utilized in optimized liposome formulations to increase cytosolic antigen release [67]. The first report of liposome-mediated vaccine adjuvants or related antigens was made by Allison and Gregoriadis [68]. Clinically approved liposomal vaccines include Epaxal, Inflexal and Mosquirix. These vaccines are referred to as virosomes because PC membrane vesicles contain proteins obtained from viruses. Antigens could be shielded from enzymatic destruction by PC membrane vesicles.

Clinically, liposomes are found useful in the treatment of cancer. Evidence reports that liposomes are largely accumulated in tumors in high amounts as compared to normal cells [58,69]. A typical example is the increase in antineoplastic activity of doxorubicin when administered as a liposome-formulated agent, accounting for their anticancer potential. More so, an increase in antitumor properties and decrease in drug toxicity properties, when plasma concentrations of vincristine were found to have increased after being administered as a liposome-based formulation, is further evidence accounting for their anticancer potential [11,70]. In antimicrobial therapy, it can be proved that the growth of micro-organisms such as bacteria is inhibited by liposomal neomycin and penicillin [54]. Another example is the significant reduction in renal and hematological toxicity of conventional Amphotericin B when engulfed in a liposome [54]. For gene therapy, it has been well reported that positively charged liposomes act as amazing human gene delivery systems [71]. Furthermore, a gene transfer liposomal product, allovectin-7tm, has proven to be efficient against metastatic melanoma, colorectal carcinoma and renal cell carcinoma, evidencing its potential in gene therapy [54].

In view of all the numerous merits and applications of liposomes, there are major challenges when it comes their long-term preservation and storage before use. Circumstances, such as the fusion of vesicles or aggregate formation, leakage of drugs, hydrolysis and/or oxidation of lipids, are some inherent challenges related to the freezing, storage and preservation of liposomal formulations [72,73,74]. Other challenges such as decreased half-life and circulation time of drugs occur upon parenteral administration of liposomes. These limitations subsequently affect the efficacy, biodistribution and safety of liposomal formulation; hence, it is imperative to devise techniques geared at optimizing liposome stability. Therefore, we examine the interactions of CPAs with the major liposomal components, such as phospholipids and cholesterol, and their thermotropic behaviors.

## 3. Phospholipid Bilayer Interactions with CPAs and Their Thermotropic Behavior

Preferential exclusion theory, which was initially put forth for proteins [75,76,77] and later for membranes [78,79], explains one of the mechanisms by which CPAs provide protection to biomolecules in cells. This theory postulates that interactions between proteins or membranes and co-solvents are thermodynamically less advantageous than interactions between proteins or membranes and water, which results in the exclusion of the co-solvent from the hydration shell enclosing these biomolecules. This increases the energy barrier required for protein denaturation in the case of proteins and stabilizes the native state [80]. The behavior of the membrane phase is modified by the preferential exclusion of co-solutes from the membrane surface. Co-solutes that are preferentially excluded cause an osmotic stress at the membrane interface, which tends to stabilize low surface area lipid phases and draw water out of multilamellar membrane stacks. Phospholipid bilayers’ interactions with various CPAs have been thoroughly investigated using both experimental methods and molecular dynamics simulation [81,82,83,84,85]. According to reports, the addition of CPAs causes the membrane to expand laterally, which reduces the thickness of the bilayer. The extent of this effect is dependent on the kind and concentration of the cryoprotectant [86]. Additionally, CPAs have an impact on the liposomes’ membrane phase transition [87,88].

Liposomes must be developed to ensure stable storage at elevated subzero temperatures while maintaining storage sufficiently below the glass transition point (*T_g_*). A glass is termed as the metastable supercooled liquid state with constrained molecular mobility [88]. Ordinary CPAs such as DMSO can be mixed with molecular compounds with high glass transition temperatures to improve the *T_g_* of formulations for use at 80 °C. Disaccharides (sucrose, lactose, trehalose) and polymers, such as Ficoll, poly-ethylene glycol (PEG), hydroxyethyl starch (HES), polyvinyl alcohol (PVA) and polyvinyl pyrrolidone (PVP), are examples of such substances [89,90,91].

Distinct phases of hydrated phospholipid membranes are distinguished by different locations (i.e., lateral order) and orientations (i.e., rotational order) [92], with the gel and fluid phases of bilayer phospholipids being the two extreme phases. Because hydrocarbon chains have an all-trans orientation that favors maximum elongation, the gel phase, also known as the solid-ordered phase, is characterized by the creation of an extremely compact bilayer with minimal mobility, as seen in Figure 5 [15]. The lipid bilayer undergoes a gel–liquid transition above a particular temperature, known as phase transition temperature (*T_m_*), in which the lipid chains change configuration (from all-trans to all-gauche) and acquire a less stretched and compact shape. As seen in Table 2, *T_m_* can vary substantially between lipids depending on their chemical structure (i.e., acyl chain length and saturation degree, polar head nature and dispersion medium type and ionic force).

Moreover, the degree of unsaturation and length of acyl chains in particular have the greatest influence [93,94]. The *T_m_* of the phospholipid increases with increasing acyl chain length (or decreasing unsaturation level) (Table 2). Sometimes, the gradual change in the separation between the polar heads (pre-transition, *T_p_*), which occurs a few degrees before *T_m_* (5–7 °C), occurs before the transition from the gel to the liquid phase. The bilayer surface is characterized by periodic one-dimensional undulations, known as the ripple phase, above the *T_p_* value, due to simultaneous variations in the phospholipid structure and membrane curvature. Since various domains have distinct geometrical properties, lipids are compelled to arrange on the surface. It is hypothesized that ripples are created by the alternating of gel and liquid lipid domains in a single monolayer [95]. According to reports, saturated acyl chains between C_10_ and C_13_ travel directly from gel to liquid, rather than passing through the ripple phase (Table 2) [96].

Furthermore, hydrocarbon chains can be tilted or not tilted depending on the hydration level; the angle of tilt increases as the water content increases, resulting in a thin bilayer [97]. Other factors that influence chain tilt include the type of phospholipid polar head and the presence of cholesterol [98]. Cholesterol, along with phospholipids, is one of the most important components of liposomal formulation because it maintains the fluidity of the bilayer and helps to stabilize the membrane. The hydrophobic steroidal moiety of cholesterol (and other similar substances) actually promotes the all-trans configuration of acyl chains in the gel phase, lowering the tilt angle. Moreso, the addition of cholesterol in the bilayers produces a broadening, or elimination, of the *T_m_* by exerting an ordering impact on the liquid phase, which takes on physical properties similar to the solid-ordered gel phase. In reality, the liquid-ordered phase describes the bilayer configuration in the presence of cholesterol [15,99].

## 4. Liposomal Response to Freezing and Lyophilization

There are three basic stages to a conventional freeze-drying process, that is: freezing, primary drying and secondary drying. The freezing phase is a cooling process where the majority of the solvent is separated from the excipients and liposomes, causing ice crystals to form. Another method for reducing sample heterogeneity and drying rate, which is mostly caused by the growth of ice crystal size, is to anneal a frozen sample [15].

### 4.1. Effect of Frezzing

Since *T_m_* is related to the hydration state of phospholipids, and liposomal dispersions can account for various types of water pools, the freezing step can result in a number of destabilizing stress factors. Bulk water and intraliposomal solution freeze at approximately −20 °C (heterogeneous ice nucleation) and −45 °C (homogeneous ice nucleation), respectively [103]. When lipid–water suspensions are highly hydrated and above freezing, they split into two distinct phases: a lamellar phase with roughly 30 water molecules per lipid and a bulk phase with practically pure water [104]. As the temperature drops, the water in the bulk begins to freeze and the liposomes move in close proximity to one another. The distance between the phospholipid head groups becomes smaller when the lamellar phase dehydrates. These occurrences cause the bilayer to expand laterally, creating a compressive stress in their plane. Depending on the bilayer composition, the outcomes could be the demixing of several components in vesicles, the creation of micelles or agglomeration [104].

Furthermore, freezing results in a cryoconcentration of the solutes in the bulk solution, which creates an osmotic gradient and results in a loss of the internal solution and a subsequent leakage of dissolved hydrophilic drugs [105]. For instance, 1,2-distearoyl-sn-glycero-3-phosphocholine (DSPC)/cholesterol liposomes with 10% lactose added resulted in reduced internal volume, synchronous bilayer invagination and self-fusion events that produced liposomes with the shape of peanuts [106]. According to one account, the osmotic shock is unrelated to liposome size [107]. The preservation of liposome structure is significantly impacted by freezing rate. Additionally, the formation of thin ice crystals and a uniform distribution of the protectant during ultrafast cooling may lessen the breakdown of the liposomal bilayer structure.

On the contrary, because water molecules can diffuse slowly across the bilayer when the solution becomes freeze-concentrated [108], a slow freezing rate lowers both the supercooling [103] and the osmotic pressure. In order to reduce the production of ice crystals in the inner aqueous compartment and prevent leakage, a slow freezing rate (lower than 0.5 K/min) may be used [108]. Additionally, it was proposed that slow freezing would: (1) give more time for recovery from deformations caused by mechanical and osmotic pressure; (2) reduce the recovery of vesicles at the glass-ice boundary, favoring their distribution in the glass matrix; and (3) alleviate the stress vectors affecting rigid bilayers. Moreso, the bilayer’s rigidity, or the composition of the lipids and the presence of cholesterol, substantially influences the effect of the freezing rate. In addition, liposomes with cholesterol in the bilayer are less likely to be harmed by an abrupt increase in the fluid phase’s order [13].

### 4.2. Effect of Drying

The potential drawbacks of drying processes primarily manifest themselves when the product is rehydrated [108]. In reality, the water molecules bonded to the polar heads of lipids exert their force on the spatial separation of phospholipids. The hydrophobic interactions between the acyl chains are stimulated by the dehydration of phospholipids, which increases the packing density of the bilayer [109]. As a result, the bilayer transitions from the hexagonal phase, where the lipid head groups surround the water channels, to the ribbon phase, where the lipid bilayers are tightly packed to create a two-dimensional lattice [110]. The hydrocarbon backbone’s tilt decreases after dehydration, another indicator of the bilayer’s greater order, which results in a sharp rise in the *T_m_* (up to 60 °C) [109]. This increase appears to be solely reliant on the characteristics of lipid polar heads or on the kind and nature of interactions that take place between polar heads and water molecules and/or between neighboring polar heads. For example, the intense intermolecular interactions between the phosphate and ammonium groups in phosphatidylethanolamines result in a sharp rise in *T_m_* upon dehydration (from 63 to 100 °C) [111].

Since the rehydration of phospholipids is linked to drug leakage, the stabilization of the liposomal structure depends on maintaining the *T_m_* at the values of the completely hydrated bilayers to prevent a gel–liquid transition during reconstitution [112]. It should be highlighted that cholesterol itself might prevent liposomes from drying out. The interactions between the acyl chains are actually reduced by cholesterol since it lowers the *T_m_*. Due to the presence of the OH group in the interfacial region, it may also interact with the polar heads of lipids by forming H-bonds. In fact, several researchers postulated that cholesterol and protective excipients may compete during their interaction with lipid polar heads [113]. In fact, it has been found that the presence of cholesterol reduces the medication leakage upon rehydration [107].

## 5. Applicable Cryoprotectants in Liposomal Freezing

Cryoprotectants have been used in liposomal formulations to enhance their functional properties and stability following freezing [114]. Cryoprotectants or CPAs are chemical substances known to cause water to melt at lower temperatures. In contexts other than cryobiology, such substances are usually referred to as “antifreeze”. Most of the time, a cryoprotectant concentration of between 5% and 15% is sufficient to allow a significant portion of isolated cells to survive freezing and thawing from liquid nitrogen temperature. As the temperature is dropped, growing ice squeezes cells into progressively smaller pockets of liquid that have not frozen. At any given temperature, these pockets become bigger with cryoprotectants present than they would be without them. Increased unfrozen cell pockets lessen damage from both mechanical freezing injury from ice and high salt concentration [115].

The same phenomenon is witnessed in liposomes since they are also membranous structures similar to most cells. There are common examples of CPAs being used, such as dimethyl sulfoxide, glycerol, ethylene glycol, propylene glycol, trehalose and sucrose, amongst others. Most of these cryoprotectants permeate lipid membranes and replace portions of their water contents [4]; whereas, polymers and carbohydrates, such as sucrose, glucose, trehalose and mannitol, do not permeate the membrane, but rather provide stability for the membrane by interacting with the polar heads [116,117]. Thus, CPAs can be categorized into two main groups: membrane permeating CPAs and nonmembrane-permeating CPAs. A literature review on the effects of CPAs on the melting point of lipids is shown in Table 3, which reveals that the effects of CPAs vary both on the type of lipid and the CPA concentration.

Permeating cryoprotectants are small compounds that can pass through lipid membranes with the purpose of attaining vitrification of the inner aqueous phase to minimize membrane dehydration and inhibit ice growth [118]. For instance, when a membrane that is at a fixed hydration temperature a little bit higher than its *T_m_* starts to have its temperature decreased to *T_m_*, transition occurs, followed by a decrease in area per lipid. However, this transition cannot occur if the intermembrane aqueous solution is vitrified in the presence of permeating CPA. Since the vitrified layer is a solid, it can withstand high mechanical stress. Glass will prevent the temperature drop required for the gel phase to form if the temperature is dropped past *T_m_*. Glass will be able to withstand an increasing compressive stress in the membrane when the temperature drops below *T_m_* [119]. In brief, if the solution is vitrified while the lipids are in the liquid crystal phase, the transition temperature will be significantly reduced, and the membranes will stay in the fluid state. There are also two more significant impacts of glass formation. Firstly, solute crystallization will not occur upon vitrification of the sample, and secondly, further dehydration will be severely restricted if the solution vitrifies.

Non-membranepermeating cryoprotectants are large molecules, typically saccharides and polymers that do not enter the inner hydrophilic core of the lipid membrane, but rather stay on the outer membrane or the polar phospholipid heads. Even though they do not enter cells, they limit the development of ice via the same processes as penetrating cryoprotectants. At any given sub-zero temperature, a vesicle will contract less in equilibrium with ice if its internal concentration is larger at temperatures above freezing. Moreso, the concentration of any existing solutes must be lowered in order to accommodate the addition of any new solute. A high quantity of non-membrane-permeating CPAs such as sugars lowers the concentration of ions needed to create a specific osmotic pressure. Thus, the existence of sugars lowers the excessive, dangerous ion concentrations [120]. The presence of non-permeable CPAs lowers the chemical potential of water via osmosis. The suction decreases with increasing osmotic term, which lowers the stress placed on the membrane [121]. Furthermore, high concentrations of non-membrane-permeating CPAs present in lipid membranes decrease the occurrence of two dehydration damaging effects; that is, they decrease the occurrence of non-lamellar phases, and they lower the transition temperature [18].

Both the permeating and non-permeating CPAs inhibit the formation of ice crystals to avoid piercing the lipid membrane, either from within or outside, respectively. Moreover, both CPAs are utilized in either freezing or the lyophilization process; however, nonmembrane-permeating CPAs are often used in lyophilization processes since they have the ability to prevent osmotic shock during the whole freeze-drying processes and rehydration of products. At the same dose, non-penetrating cryoprotectants are typically less harmful than penetrating cryoprotectants.

### 5.1. Dimethyl Sulfoxide (DMSO)

DMSO is a well-known permeable CPA that is commonly employed in cell cryopreservation [16]. DMSO was shown to have good permeability to living cells and could protect frozen red blood cells and bull sperm cells after its discovery in 1959 [122]. Quite a number of studies have shown the role of DMSO as an important CPA. A study conducted by Sydykov et al. [20] showed that DMSO was able to prevent the CF-leakage in the freeze-thaw process of liposomes. Additionally, in order to avoid the high cost involved in the use of liquid nitrogen, a group of researchers decided to store liposomes in a mechanical freezer at −80 °C and allow its shipment on dry ice. They looked into the idea of utilizing a combination of DMSO and sucrose to raise the storage temperature of cryopreserved liposomes to −80 °C and determined the preservation efficacy by measuring the stability of liposomes loaded with carboxyfluorescein (CF) for 3 months at various storage temperatures (−25 °C, −80 °C, and −150 °C). It was discovered that the CF-leakage rate of liposome samples stored at −80 °C was very minimal and negligible, despite the fact it was above or near to the *T_g_* for the DMSO/sucrose formulations. In contrast, it was discovered that 10% DMSO is favored over 5% DMSO in terms of liposome leakage at −25 °C, where the rates of liposome leakage were higher in the presence of 1 M sucrose than those in the presence of 0.5 M sucrose [88].

Another study investigated the membrane phase behavior of liposomal dispersions with DMSO with liposomes made of dimyristoylphosphatidylcholine. The effects of DMSO were examined both within and around the liposome, and the differential scanning calorimetry (DSC) was also used to examine the phase transition temperatures. According to their findings, DMSO significantly affected the phase behavior of liposomal dispersions, particularly by lowering the freezing point of the intraliposomal medium, raising the lipid’s main phase transition temperature and improving the structural integrity of freeze-thawed liposomes as concentrations of DMSO in the dispersion’s increase [123].

Furthermore, lipid suspensions in DMSO/water solutions between −60 to 30 °C were examined using Fourier transform infrared spectroscopy. The effect of the solvents on the thermotropic ad structural behavior of cholesterol-loaded liposomes (POPC/chol) was investigated. Upon examination of the characteristics of liposomes suspended in water and a DMSO/water solution with a 0.10 DMSO mole fraction, it was found that DMSO addition increases the thermal stability of the membrane’s gel phase. It was then predicted that the inclusion of DMSO, both in the gel and liquid-ordered phases of the membrane, significantly reduced the amount of unfrozen water at −60 °C [124]. Intriguingly, this study also exhibited that DMSO reduced the hydration of the lipid heads when the lipid vesicles were dispersed in a water solvent, but it had no effect on the hydration of the phosphate and carbonyl groups when the membrane was frozen.

So, it is evident that DMSO lowers the electrolytic concentration in the remaining cooled fluids surrounding and within liposomes at any given temperature. Nonetheless, DMSO has some disadvantages that limit its therapeutic use such as its inherent toxicity [125,126] and the time-consuming washing processes required to reduce its negative effects [127].

### 5.2. Glycerol

Glycerol is a clear, odorless liquid that is a simple polyol or sugar alcohol molecule, which forms hydrogen bonds with water molecules thanks to its favorable kosmotropic characteristics [128]. Due to this circumstance, it is challenging for a mixture of 70% glycerol and 30% water to form ice crystals. Glycerol is less harmful at high concentrations as compared to other cryoprotectants [115] and can protect lipid membranes upon dehydration.

For instance, a study was conducted to comprehend the ability of glycerol to protect dehydration of lipid membrane as a result of osmotic stress. In order to examine the molecular mechanism underlying the protective function that regulate the solid to liquid phase transition in the phospholipid bilayers, two models of liposome samples (dimyristoylphosphatidylcholine (DMPC)–glycerol–water and DMPC–urea–water) were employed. It was shown that glycerol and urea both stabilize liquid crystalline bilayers at relatively low humidities (down to 75% RH at 27 °C), but a solid gel phase is induced in the pure DMPC–water system at 93% RH. This indicated how glycerol and urea can help to defend against osmotic stress. Furthermore, it was found that the solvent volume, not the composition, determined the phase behavior for lipid systems with restricted access to solvent [83].

Additionally, glycerol is used in the preparation of cosmetic liposomes to replace water to enhance the penetration of active cosmetic agents into the epidermis. Upon numerous critical examinations of the glycerol-based liposomes, it was observed that they possessed microbiological and physical stability [129].

In another study conducted by Marín-Peñalver et al., in both the presence and absence of glycerol, a collagen hydrolysate (HC) extracted from the tunics of a giant squid (Dosidicus gigas) was encapsulated in soy phosphatidylcholine liposomes. In order to compare the effects of adding glycerol, they were either directly added to the film-forming dispersion or into already formed liposomes. From transmission electronic microscopy, it was observed that the liposomes in the films were intact at cryonic temperatures due to reduced water solubility. In addition, the outcomes showed that glycerol-containing liposomes were less impacted by the drying of the film and the subsequent simulating of gastrointestinal digestion, with greater preservation of vesicle size and morphology, than if glycerol was introduced straight to the film-forming fluid. [130].

To test the ability glycerol and other carbohydrates to affect the stability of liposomes, the particle size, the lipid bilayer thickness and lamellarity were examined with the aid of photon correlation spectroscopy and small angle X-ray scattering [14]. Additionally, differential scanning calorimetry was used to assess the effect of cryoprotectants on the thermal lipid phase behavior of either lyophilized/rehydrated PEGylated or frozen/thawed liposome formulations. It was found that, regardless of the type of carbohydrate utilized, a mixture of glycerol and carbohydrate concentrations of about 1% (*w*/*v*) produced the best results for maintaining the average size of the extruded unilamellar liposomes after freezing. The bilayer organization showed no appreciable modifications, and the effects of freezing on lipid transition behavior were minimal. Similar carbohydrate amounts to those used for freezing were sufficient in the case of freeze-drying to retain the size of PEGylated liposomes upon reconstitution of the dried lyophilized cakes.

### 5.3. Sugars and Disaccharides

Despite the fact that disaccharides or sugars are the most commonly used CPAs next to DMSO, their mechanisms of action are not yet fully unraveled. It is postulated to have a combination to two mechanisms, which are the water replacement and vitrification mechanisms. The water replacement mechanism, which was proposed by Crowe et al. in 1973 [131], is based on the potential of carbohydrates to form hydrogen bonds between three phospholipids and replace the bonds with water molecules to maintain the structure of liposomes [132]. More precisely, these sugars can form H-bonds with both the carbonyl and phosphate groups of the polar heads, as well as the methyl group of the hydrophobic moiety. Despite this, the phosphate group is preferred for interactions [107,131]. Sugars reduce van del Waals interactions between hydrocarbon chains by increasing the distance between phospholipid polar heads (Figure 6). This, as a result, does not only reduce *T_m_* during dehydration, but also lower *T_m_* in fully hydrated bilayers [107,108].

The other model used to better explain the protective effect of carbohydrates is the vitrification model. In this model, it is suggested that sugars produce an amorphous phase with high viscosity and mobility upon freezing, which functions as a barrier between bilayers. This glassy matrix prevents vesicle fusion and shields the bilayer from ice formation damage [133]. Additionally, sugars help lipids with their *T_m_* and prevent hydrophilic compounds’ leakage caused by extra-liposomal ice. To further support the vitrification theory’s implications in the protection of liposomes by sugars, it was postulated that the surface tension of vesicles reduces as a result of interactions between the liposome surface and the glassy matrix [112,134]. For instance, paclitaxel retention was assured in paclitaxel-loaded pegylated liposomes upon addition of sucrose at a sucrose concentration of 150 mM (sugar:lipid ratio 3:1 *w*/*w*), with the maximum protectant effect observed at a 5:1 sugar:lipid ratio in preventing aggregation and assuring the presence of a mono disperse population.

Furthermore, the investigation of an appropriate freeze-drying formulation for liposomes was investigated by Susa et al. After the freeze-thaw cycles were carried out, it was seen than the disaccharides (cellobiose, glucose, lactose, sucrose and trehalose), either alone or in combination, reduced the osmotic stress, stabilized the liposome and eventually protected the integrity of the liposomes [107].

When liposomes are frozen, the excipient keeps the size of the liposomes constant, while also reducing the osmotic gradient caused by cryo-concentration [135]. On the contrary, the excipient stabilizes the lipid bilayers in the liposomes’ outer compartment, limiting changes in their physical characteristics and eventual drug leakage [15].

Recently, novel ciprofloxacin nanocrystals inside liposomes (CNL) powder formulations for controlled release and inhalation were created. The storage stability of CNL powders containing ciprofloxacin (CIP), lipids, lyoprotectant (such as sucrose or lactose), magnesium stearate or isoleucine was examined in the study [136]. These powders were made by spray drying, collected in a dry box with a relative humidity (RH) of around 15%, and then kept at room temperature with either a 4 or 20%RH. Over a six-month period, the stability of liposomes, CIP encapsulation efficiency (EE), aerosol performance, in vitro drug release (IVR) and solid-state properties were assessed. Over six months of storage at 4%RH, sucrose CNL powder maintained continuous aerosol performance, liposomal integrity and regulated release of CIP. However, after being stored at 20%RH for the same amount of time, sucrose crystallized, which significantly reduced EE and aerosol performance (*p*-values 0.05) and led the IVR of CIP to converge with that of the non-crystalline CIP liposomal dispersions (f2 > 50). Regardless of the storage RH, lactose CNL maintained exceptional aerosol performance over the course of six months. However, within the first month of storage, both RHs experienced liposomal instability with a significant decrease in EE and an increase in liposome size (*p*-values 0.05). In addition, regardless of the storage RHs, the IVR assay of CIP from lactose CNL revealed a less regulated release and a significant difference (f2 < 50) from its initial value after six months [136]. From the study we can see that dry powder inhalers of CNL containing sucrose when kept below 4%RH at room temperature for six months, possess more physiochemical stability compared to those containing lactose. Additionally, we observed that both protectants cannot stabilize liposomes at high RHs (>20%). Therefore, it is best for them to be stored at low RHs to gain optimal stability.

Water replacement and vitrification theories are widely acknowledged to work together to maintain liposomes during freeze-drying rather than being mutually exclusive [137]. They are unable to fully explain the findings reported in the literature, though. In the presence of highly concentrated solutes, it should also be taken into account that the Tm of dispersions rises as the water activity falls [18]. For instance, Strauss et al. discovered that the *T_m_* was raised by several degrees when up to 10% sucrose was added to hydrated multilamellar vesicles of DPPC [138]. Additionally, the kind of vesicles and the degree of protectant affect the system’s thermal behavior. When mono and disaccharides were added, the Tm was elevated and widened in the case of large DPPC multilamellar liposomes. Multiple metaphases were produced when large concentrations of trehalose and sucrose were added to unilamellar vesicles formed of the same phospholipid [139]. Furthermore, it is widely accepted that high sugar concentrations, which are necessary to ensure a repeatable protective effect, may affect the viscosity and, thus, the rehydration of the vesicles [140]. To prevent having an adverse impact on the safety of the drug product supplied by ophthalmic and parenteral route, its impact on the tonicity of the reconstituted solution must be carefully considered when the amount of protectant is defined.

In other cases, a method for integrating protectants into a single formulation is necessary to allow lyoprotectants to increase a formulation *T_g_* while also offering direct interactions with lipid bilayers. With egg phosphatidylcholine liposomes after freeze drying, mixtures of phosphate anion and sucrose were demonstrated to improve liposomal solute retention to 85%, compared with full leakage using phosphate alone and 75% retention with sucrose alone. By forming hydrogen bonding networks between phosphate and sugar molecules, the excipient mixture significantly raised the formulation *T_g_* [141]. Combining glucose with hydroxyethyl starch has been shown to stabilize liposomes; however, when used independently, glucose is unable to prevent liposome fusion and hydroxyethyl starch is unable to prevent solute leakage during drying [142]. It has also been demonstrated that mannitol and trehalose work well to preserve liposome dispersions. A lyoprotectant composition for lyophilized liposomes that combines cyclodextrin and a saccharide is also disclosed in various patents [143]. Mannitol by itself does not offer much protection due to its tendency to crystallize during freeze drying, but when combined with glucose, the resulting liposome drug retention increased to 86.5%.

Even though the individual of saccharides and sugars are efficient in preserving and stabilizing liposomes, it is evident that a combination of their use is more effective and provides a much more stable liposomal formulation. Additionally, because the bulk of liposomes are preserved by freeze-drying, carbohydrates and sugars have been observed to be the most commonly employed CPA in liposome preservation.

### 5.4. Polyampholytes

A new crop of polymeric cryoprotectants have evolved which are the poly-ampholytes. These poly-ampholytes have charged polymers with both positively and negatively charged groups [144], for example, carboxylated polylysine. They interact with membranes and have potential as a cryoprotectant in cryovial vitrification, monolayer slow-vitrification and cryovial slow-freezing. Currently, finding innovative polyampholytes to use as CPAs is also a prominent area of research. For instance, a new polyampholyte has been made by polymerizing poly(methyl vinyl ether-alt-maleic anhydride) with dimethylaminoethanol [145].

In a study to determine the ability of polyampholytes to avoid drug leakage, Rajan et al. developed a polyampholyte CPA from copolymer of 2-(dimethylamino) ethyl methacrylate (DMAEMA) and methacrylic acid (MAA) (poly(MAA-DMAEMA)) [146]. The leakage of soluble marker, CF, decreased upon addition of poly (MAA-DMAEMA) to the liposomal formulation. The polymer proved to protect the membrane during the freeze-thaw process, and that cryoprotection increased as the concentration of polymer increased.

Another study employed the use of reversible addition-fragmentation chain transfer polymerization to develop a completely synthetic polyampholyte. This polyampholyte showed to be non-cytotoxic and also protected liposomal membrane against leakage during freezing and stabilized them [147]. Just like other polymers and cryoprotectants, polyampholytes prove to be promising and competent in stabilizing liposomal formulations. However, unlike sugar and carbohydrates, there seems to be not much information on their application in the preservation and stabilization of liposomes. Therefore, we propose that some research attention be dedicated in that direction in order to thoroughly examine and comprehend their usage to add to and expand the pool of CPAs suited for liposomal preservation and stabilization.

## 6. Conclusions and Prospects

Liposomes undergo freezing during lyophilization and freeze-thawing, which are done to prevent the leakage of constituents and instability of liposomes, thereby enhancing their application as drug delivery systems. In the presence of cryoprotectants, while excluding solutes and particles into a cryo-concentrated liquid phase, water freezes into ice crystals. When in equilibrium, the liquid phase concentration might adhere to the equilibrium phase boundary (equilibrium freezing curve). Subsequently, the liquid phase reaches the maximal cryo-concentrated solution’s glass transition temperature [148].

When a cryoprotectant reaches a certain temperature, it vitrifies, creating a glassy matrix that can shield nanoparticles from the ice’s mechanical stress. The majority of research on cryoprotectants show their inclusion improves liposome stability [149]. In essence, the efficacy of freeze-drying to maintain liposome stability in an aqueous state depends on a complex relationship between protectants and bilayer composition, not merely the kind and position (the inner core or bulk water) of the protectant or the determination of process parameters.

Nonetheless, several types of CPAs exhibit varying efficient mechanisms of action in freezing of liposomes; therefore, the combination of different types of cryoprotectants might be worthwhile and innovative due to the synergistic effects of the CPAs. Furthermore, the adequate use of permeating and noncytotoxic cryoprotectants such as osmoprotectants in place of DMSO will be a more prospective and forward-looking approach to obtain a safe freezing process. Unlike DMSO, osmoprotectants have no toxicity, and lipid membranes may rapidly import osmoprotectants to reverse water outflow and prevent osmotic damage without disrupting vital activities in a hypertonic environment [150]. Osmoprotectants should be able to protect liposomes from osmotic damage during freezing, and due to their hydrophilic molecular nature, they should have the ability to inhibit ice formation and hence protect lipid membranes from ice injury [151]. Some examples of natural biocompatible osmoprotectants are mostly amino acids such as proline, glycine and taurine.

The mechanical stresses that liposomes frequently experience during the freezing process can have an impact on the end product’s quality and safety. By incorporating the right concentration of CPAs under the right conditions, these problems can be avoided. In summary, finding the right CPAs for effective and successful freezing of liposomes can help reduce the lengthy process of lyophilization process, which later would require the resuspension of liposomes again in an aqueous form before administering. Our review reveals that DMSO, sugars and carbohydrates are the most commonly utilized cryoprotectants in liposome freezing. Nonetheless, recent cryopreservation research has led to the discovery of a wide range of substances with potent cryoprotection properties including antifreeze proteins and metallic frameworks such as nanoparticles. It is therefore imperative that researchers explore these novel materials in liposomal formulation, preservation and stabilization.

## Figures and Tables

**Figure 1 ijms-23-12487-f001:**
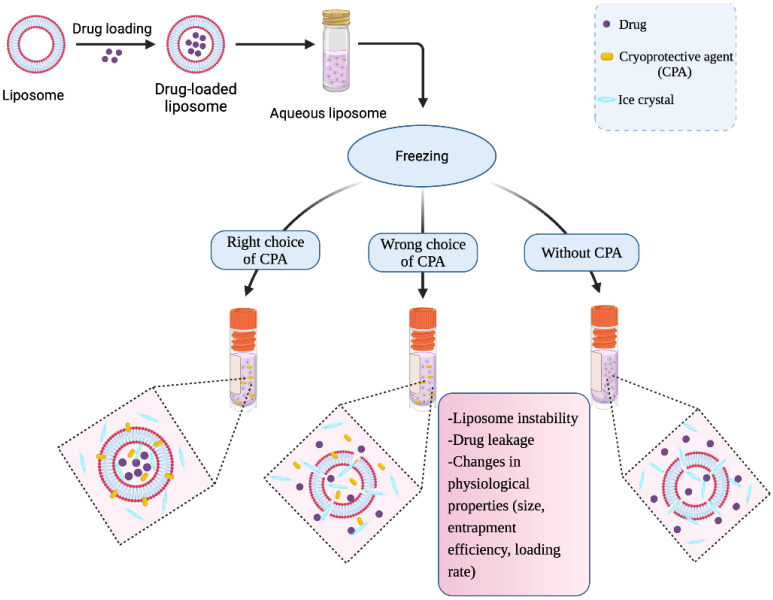
Freezing/lyophilization of drug-loaded liposomes with and without the addition of cryoprotective agents (CPAs).

**Figure 2 ijms-23-12487-f002:**
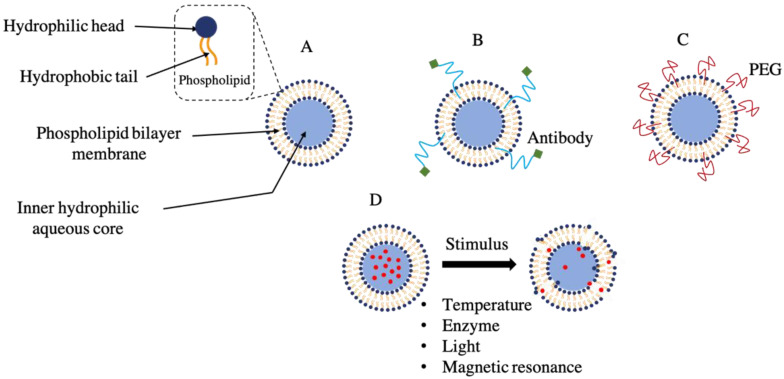
Classification of liposomes based on their composition. (**A**) Conventional liposome, (**B**) pH sensitive liposome, (**C**) long circulating liposome, (**D**) temperature-sensitive, enzyme-sensitive, light-sensitive, and magnetic-response liposomes.

**Figure 3 ijms-23-12487-f003:**
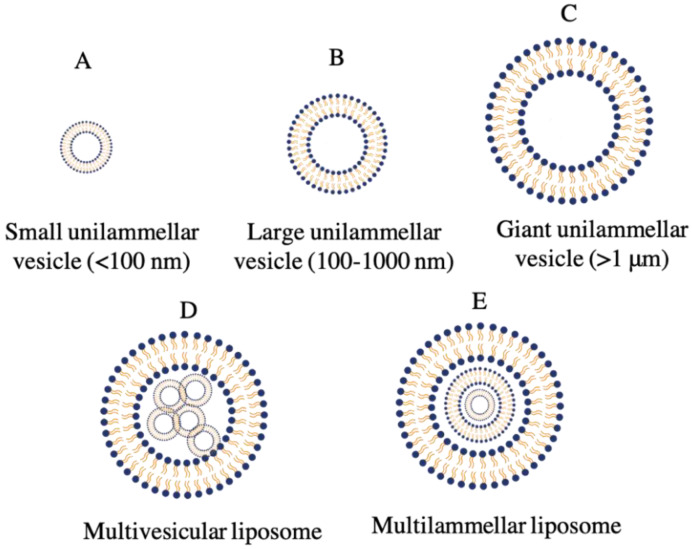
Classification of liposomes based on their lipid bilayer structure and size. (**A**) Small Unilammelar Vesicles (SUV), (**B**) Large Unilammelar Vesicles (LUV), (**C**) Giant Unilammelar Vesicles (GULV), (**D**) Multivesicular Liposome, (**E**) Multilammelar Liposome.

**Figure 4 ijms-23-12487-f004:**
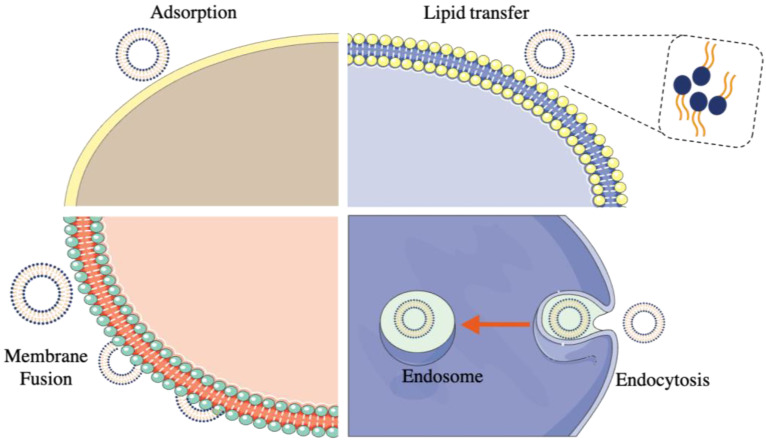
An illustration of the interactions between liposomes and cells.

**Figure 5 ijms-23-12487-f005:**
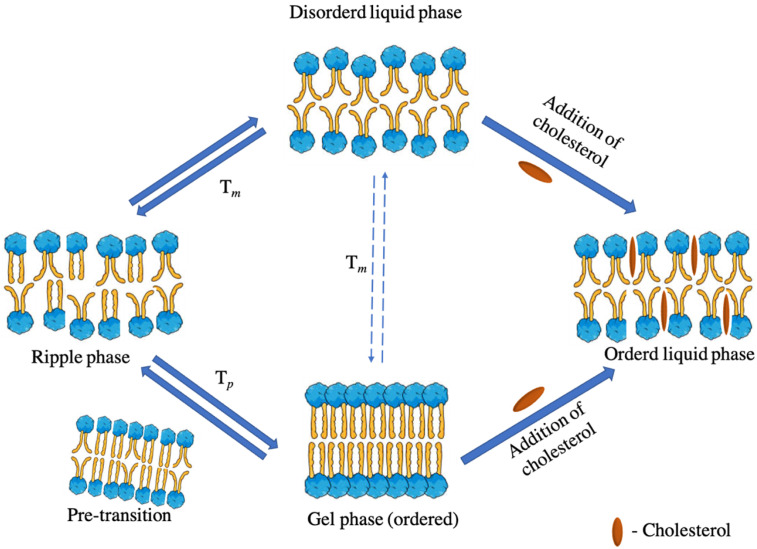
An illustration of the thermotropic behavior of hydrated phospholipids in both phase transition temperature (*T_m_*) and pre-transition temperature (*T_p_*).

**Figure 6 ijms-23-12487-f006:**
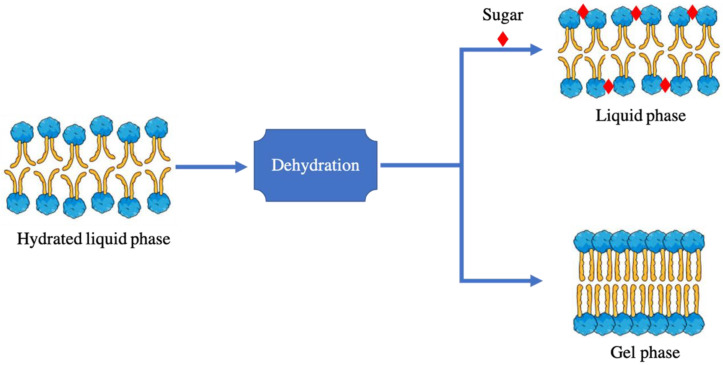
The effect of sugar to phospholipid bilayers before and after dehydration.

**Table 1 ijms-23-12487-t001:** A table indicating typical FDA approved and marketed liposomal products [6].

Active Product	Trade Name	Lipid Composition	Formulation Type	Targeted Disease	Company
Amphotericin B	Ambisome	HSPC:DSPG, chol 2:0.8:1 M	Freeze dried	Fungal and protozoal infection	Gilead Sciences
Amphotericin B	Amphotec	Cholesteryl sulphate:Amphotericin B 1:1 M	Freeze dried	Severe fungal infections	Ben Venue
Amphotericin B	Abelcet	DMPC:DMPG 7:3 M	Aqueous dispersion	Invasive severe fungal infections	Enzon
Amykacin	Arikayce	DPPC:chol	Aqueous dispersion	Mycobacterium avium lung disease	Insmed
Cytarabine	Depocyte	DOPC:DPPG	Aqueous dispersion	Malignant lymphomatous meningitis	Pacira (formerly Skye Pharma)
Daunorubicin	DaunoXome	DSPC:chol 2:1 M	Aqueous dispersion	HIV-related Kaposi’s sarcoma	Gilead Sciences
Daunorubicin/Cytarabine	Vyxeos	DSPC:DSPG:chol 7:2:1	Freeze dried	Therapy-related acute myeloid leukemia (t-AML) or AML with myelodysplasia-related changes (AML-MRC)	Jazz
Doxorubicin	Myocet	EPC:chol 55:45 M	Freeze dried	Combination therapy with cyclophosphamide in metastatic breast cancer	Zeneus
Glycoprotein E based vaccine	Shingrix	AS01b:MPL-L; QS-21 (n), DOPC, chol	Aqueous dispersion	Vaccine for the prevention of shingles (herpes zoster)	GSK
Inactivated hepatitis A virus	Epaxal	DOPC:DOPE 75:25 M	Aqueous dispersion	Hepatitis A	Berna Biotech
Inactivated hemagglutinin of Influenza virus strains A and B	Inflexal V	DOPC:DOPE 75:25 M	Aqueous dispersion	Influenza	Berna Biotech
Irinotecan	Onivyde	DSPC:MPEG-2000:DSPE 3:2:0.015 M		Combination therapy with fluorouracil and leucovorin in metastatic adenocarcinoma of the pancreas	Merrimack Pharmaceuticals
Morphine Sulphate	DepoDur	DOPC, DPPG, Cholesterol and Triolein		Pain management	SkyPharma
Paclitaxel	Abraxane		Freeze dried	Non-small-cell lung cancer (NSCLC), metastatic breast cancer and pancreatic cancer	Abraxis BioScience
PEG-doxorubicin	Doxil/Caelyx	HSPC:chol:DSPE-PEG 56:39:5 M	Aqueous dispersion	Metastatic ovarian cancer, AIDS-related Kaposi’s Sarcoma, multiple myeloma	Ortho Biotech, Schering-Plough
Verteporfin	Visudyne	EPG:DMPC 3:5 M	Freeze dried	Ocular histoplasmosis, age-related macular degeneration, pathologic myopia	QLT, Novartis

**Table 2 ijms-23-12487-t002:** A table showing the increasing acyl chains with increasing phase transition temperature (*T_m_*).

Phospholipids	Pre-Transition Temperature (*T_p_*/°C)	Transition Temperature (*T_m_*/°C)	Acyl Chains	Ref.
DPPC	35.5	40.5	16:0/16:0	[100]
DOPC	9.0	−18.0	18:1	[96]
DSPC	54.5	49.1	18:0	[96]
DMPC	22.0	24.0	14:0/14:0	[96]
DLPC	-	−1.0	12:0/12:0	[96,101]
EPC	-	−15.0 to −20.0	Mixed chains	[96]
HSPC	47.8	53.6	16:0/18:0	[102]

1,2-dipalmitoyl-sn-glycero-3-phosphocholine (DPPC); 1,2-dioleoyl-sn-glycero-3-phosphocholine (DOPC); 1,2-distearoyl-sn-glycero-3-phosphocholine (DSPC); 1,2-dimyristoyl-sn-glycero-3-phosphocholine (DMPC); 1,2-dilauroyl-sn-glycero-3-phosphocholine (DLPC); egg phosphocholine (EPC); hydrogenated soy phosphocholine (HSEP).

**Table 3 ijms-23-12487-t003:** Effects of CPA concentrations on membrane phase transition temperatures of lipids [20].

Phospholipids	CPAs	CPA Ratio	Transition Temperature (*T_m_*/°C)
DPPC	DMSO	0.05 molar	42.5–44.5
	DMSO	0.9 molar	57.8, 58.3
	GLY	5% wt	41.7
	EG	20% *w*/*v*	40.7
	EG	55% *w*/*v*	42
	acetone	50% *v*/*v*	37.3
DSPC	DMSO	14% *w*/*v*	55.5
	PG	70% *w*/*v*, anhydrous	48.5
DMPC	DMSO	35% wt	29.9
	GLY	40% wt	24.6
DMPE	DMSO	0.15 molar	56.5
	GLY	0.17 molar	60
DOPA	EG	50% *v*/*v*	−11
DOPC	EG	50% *v*/*v*	−14

Dimethyl sulfoxide (DMSO); glycerol (GLY); ethylene glycol (EG); propylene glycol (PG); 1,2-dipalmitoyl-sn-glycero-3-phosphocholine (DPPC); 1,2-dioleoyl-sn-glycero-3-phosphocholine (DOPC); 1,2-distearoyl-sn-glycero-3-phosphocholine (DSPC); 1,2-dimyristoyl-sn-glycero-3-phosphocholine (DMPC); 1,2-dimyristoyl-sn-glycero-3-phosphoethanolamine (DMPE); 1,2-dioleoyl-sn-glycero-3-phosphate (DOPA).

## Data Availability

All data used to support the findings of this review are included within the article.

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
