# Peer review of "The Role of Cryoprotective Agents in Liposome Stabilization and Preservation"

_ijms, 2022, doi:10.3390/ijms232012487_

Round 1
Reviewer 1 Report
The review by Boafo et al. titled “The Role of Cryoprotective Agents in Liposome Stabilization
and Preservation” aims to analyze the use of liposomes as drug delivery vehicles, the interactions between CPAs and lipids and their thermotropic behavior, and some examples of CPAs for liposomes freezing.
First of all, the topic dealt from the review seems not to be in line with the special issue “Biophysics in Membrane of Cells”. It requires “to reveal the biophysics in the membrane of cells, studying the structure and function of the membrane, the molecular mechanisms of various activities of the cell, the dynamic understanding of membranes, the role of lipids in membranes, the structure of channels as well as their opening and closing processes, the structure of receptors and their specific interactions with ligands, the information transfer mechanism, the constituent structure of electron transfer chains and their motion, and energy conversion mechanisms”. Since this review is about liposomes and preservation, and, since interactions with membrane of cells is completely not investigated, in my opinion, this review does not fit the theme of this special issue.
Regarding the review, the treated topic is superficially and little investigated. The application of liposomes is scarcely deepened, for instance, no mention has been made about the most recent applications of liposomes in the vaccine field, the current FDA approved liposomes (and already used in therapy), such as Doxil, Abraxane, and other have not even been cited. In addition, the structure, the composition, the types, and the production method of liposomes are totally not examined, even though the peculiar structure of liposome is a crucial aspect in their preservation. The presence of two aqueous phases, an inner and an outer one, and the bilayer composition, orientation and structure are significant parameters that must be considered when talking about preservation, stability, and integrity of liposomes after a freezing or freeze-drying process.
Considering the CPAs, the different types, penetrating and not penetrating ones, and the theories regarding their mechanisms of protection are mentioned several times, but not clearly explained. The examples are few and the distinction between the CPAs using for freezing and freeze-drying is not clear.
In the main text, there is a reference to a Table 1 (line 78), which is not present in the manuscript, and the numbering of the tables started for Table 2.
In general, the manuscript is not well organized and the discussion hasty.
Reviewer 2 Report
An editing error and two typos are found (to be fixed).
o Page 3 (2nd para)
(i) (L 78) Table 1 is missing (should be added)
(ii) (L 81) a typo... systematically (systemically)
o Page 8 (3rd para)
(i) (L 306) a typo... after (After)
(1) Liposomal formulation affect clincal efficacy of drugs, and the numerous of them are already on the market or thorough clinical trials.
(2) The authors present an array of basic cryo-protective agents and their methods for formulating & preserving liposomal drug carriers.
(3) In order to overcome the problems of drugs with poor pharmacokinetics and/or limited bioavailability, new methodologies of formulation/preservation of drug carriers are as important as finding new drugs, especially in searching for brain-targetting anticancer drugs to evade the brain-blood barrier.
(4) I think the information found in this review and the references there-in are very helpful to understand the basic issues of lipid formulations, such as the types of CPAs, phase transition, and interaction with drugs, and to develop better carriers to pass, effectively & non-toxically, through the cell membrane.
